# Attosecond Transient Absorption Below the Excited States

**Jinxing Xue [1,2], Xinliang Wang [3], Meng Wang [4,*], Cangtao Zhou [2] and Shuangchen Ruan [2]**

1    College of Physics and Optoelectronic Engineering, Shenzhen University, Shenzhen 518060, China; xuejinxing@sztu.edu.cn
2    Shenzhen Key Laboratory of Ultraintense Laser and Advanced Material Technology, Center for Advanced Material Diagnostic Technology, Shenzhen Technology University, Shenzhen 518118, China; zcangtao@sztu.edu.cn (C.Z.); scruan@sztu.edu.cn (S.R.)
3    State Key Laboratory of High Field Laser Physics and CAS Center for Excellence in Ultra-Intense Laser Science, Shanghai Institute of Optics and Fine Mechanics, Chinese Academy of Sciences, Shanghai 201800, China; wxl@siom.ac.cn
4    Sino-German College of Intelligent Manufacturing, Shenzhen Technology University, Shenzhen 518118, China
*    Correspondence: wangmeng@sztu.edu.cn

**Abstract:** In this study, the attosecond transient absorption (ATA) spectrum below the excited states of the helium atom was investigated by numerically solving the fully three-dimensional time-dependent Schrödinger equation. Under single-active electron approximation, the helium atom was illuminated by a combined field comprising of extreme ultraviolet (XUV) and delayed infrared (IR) fields. The response function demonstrates that the absorption near the central frequency ($\omega_X$) of the XUV field is periodically modulated during the overlapping between the XUV and IR pulses. Using the time-dependent perturbation, the absorption near $\omega_X$ is attributed to the wavepacket excited by the XUV pulse. The wave function oscillating at the frequency of the XUV pulse was obtained. Furthermore, the chirp-dependent absorption spectrum near $\omega_X$ potentially provides an all-optical method for characterizing the attosecond pulse duration. Finally, these results can extend to other systems, such as solids or liquids, indicating a potential for application in photonic devices, and they may be meaningful for quantum manipulation.

**Keywords:** attosecond pulse; attosecond transient absorption; high order harmonics

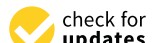



## 1. Introduction

Attosecond transient absorption spectroscopy (ATAS) is a newly developed technology to probe the ultrafast dynamics of electrons under the action of an isolated attosecond pulse from high harmonic generation and a synchronized infrared pulse (IR) [1–14]. Typically, the attosecond pulse is in the extreme ultraviolet (XUV) frequency range, i.e., between 20 and 100 eV [15–17]. The unique advantage of the ATAS is that it can simultaneously have high temporal (in attosecond scale) and spectral (limited by the soft X-ray spectrometer) resolution without violating the Fourier theory [18]. The temporal resolution is indirectly obtained by scanning the time delay between the IR and the XUV pulse; thus, the ATAS is essentially a pump-probe scheme [15]. A variety of ultrafast dynamics have been successfully investigated in atoms [1,3,4,6–9,19–23], molecules [24], or solids [10,12,25–27], indicating the universality of this spectroscopic technology.

The majority of the current ATAS methods use an isolated XUV pulse with a wide spectral bandwidth covering many intrinsic atomic and molecular levels, and previously unobserved ultrafast dynamics have been discovered, such as the light-induced-state [28–34], the sub-cycle AC Stark shift [28], Autler–Townes splitting [35], hyperbolic sidebands [15,29,32,33], and quantum beats [8,36,37]. However, the dynamics for the case where the central photon energy of the isolated XUV pulse is below the first excited state has been largely ignored. Moreover, the existence of the ultrafast dynamics below the excited states requires further exploration. Here, we focus on the absorption dynamics

below the first excited state of the helium atom under the single-active electron (SAE) approximation [38,39]; i.e., the electronic structure and electronic correlations are neglected.

For this purpose, a fully three-dimensional time-dependent Schrödinger equation (3D-TDSE) and a three-level model were used to calculate the attosecond transient absorption (ATA) spectrum. The absorption corresponding to $\omega_X$ was attributed to the wavepacket excited by the XUV field, which demonstrates the transition and coupling processes of the ATA spectrum. Using the time-dependent perturbation, the wave function of a three-level atom excited by a single XUV field was analytically obtained. In addition, the chirp-dependency of the ATA spectrum below the excited states might provide an all-optical method for characterizing the attosecond XUV pulse duration. The attosecond dynamics observed in this study may also exist in other systems, such as solids, liquids, and semiconductors, indicating widespread applicability in photonic devices.

## 2. Theoretical Model

The ATA spectrum of helium atoms was calculated by solving the 3D-TDSE in the presence of IR and XUV pulses. This 3D-TDSE has been widely used to investigate the ultrafast dynamics of electrons in atoms. The ATA spectrum of the helium atom was obtained through the response function [15]: $S(\Omega, t_d) = -2\Im[d(\Omega, t_d)E^*(\Omega, t_d)]$, where $t_d$ is the time delay between the IR and XUV pulses, and $E(\Omega, t_d)$ is the Fourier transform of the time-dependent total electric field $E(t)$. The dipole spectrum $d(\Omega, t_d)$ was obtained through the Ehrenfest theorem, where $d(\Omega, t_d) = a(\Omega, t_d)/\Omega^2$. $a(\Omega, t_d)$ is the Fourier spectrum of the time-dependent dipole acceleration $a(t, t_d)$. This dipole acceleration is constructed by $a(t, t_d) = \langle \Psi(t)|\partial H/\partial t|\Psi(t)\rangle$, where H and $|\Psi(t)\rangle$ are the Hamiltonian and wavefunction of the electron, respectively. The wavefunction is obtained by numerically solving the 3D-TDSE. A detailed description of the 3D-TDSE solution can be found in [40]. The response function $S(\Omega, t_d)$ was derived from the energy exchange between the helium atom and the total electric field; thus, the positive (negative) values represent the absorption (emission) at frequency $\Omega$. Note that $\Omega S(\Omega, t_d)$ is the energy absorbed (emitted) at $\Omega$. The IR and the XUV pulses used in the 3D-TDSE simulation have a chirped formula, and the vector potential is expressed as follows [41]:

$$A_j(t - t_d) = Re\left\{ -i\frac{1}{\omega_j}\sqrt{\frac{I_j}{1 - i\xi_j}} \exp\left\{-i[\omega_j(t - t_d) - 2\ln 2\frac{(t - t_d)^2}{\tau_j^2(1 - i\xi_j)}]\right\}\right\}, \quad (1)$$

where $j$ represents IR or XUV; the total electric field is expressed as $E(t) = -\partial A(t)/\partial t$, $A(t) = A_X(t - t_d) + A_I(t)$; $\xi_j$ is a dimensionless parameter for the chirp; and $I_j$, $\omega_j$, and $\tau_j$ are the intensity, angular frequency, and Fourier-limited (chirp free) duration of the corresponding electric field, respectively. Both the XUV and the IR pulses are linearly polarized in the same direction. The duration of the chirped pulse is $\tau = \tau_0\sqrt{1 + \xi_j^2}$. In the numerical simulation, $I_X = 1 \times 10^{10}$ W/cm$^2$, $\omega_X = 14$ eV, and $\tau_X = 500$ as. The central frequency $\omega_X$ was set to 14 eV, which is below the first excited state (2 s; 20.24 eV) of the helium atom. Such an XUV frequency is the essential in obtaining the simulation results in this study. For the IR pulse, $I_I = 1 \times 10^{13}$ W/cm$^2$, $\lambda_I = 800$ nm, and $\tau_I = 5$ fs. The duration of the IR pulse was measured at the full width at the half maximum value of the IR intensity, which is approximately $2T$ of the 800 nm pulse, where $T$ is the optical cycle (O.C.) of the IR pulse. The ionization probability of the helium atom interacting with the IR pulse can be safely ignored.

As all physical effects are included in the 3D-TDSE, it is impossible to extract specific processes from its results. Thus, a three-level model was applied to capture the essential features of the ATA spectrum [29,31–33]. In this model, the wavefunction was expanded into the finite set of atomic states: $|\Psi(t)\rangle = c_1(t)e^{-i\epsilon_1 t}|1s\rangle + c_2(t)e^{-i\epsilon_2 t}|2s\rangle + c_3(t)e^{-i\epsilon_3 t}|2p\rangle$, where $c_1(t)$, $c_2(t)$, and $c_3(t)$ are the time-dependent coefficients of the 1s, 2s, and 2p eigenstates, respectively. $\epsilon_1$, $\epsilon_2$, and $\epsilon_3$ are the eigen energy of 1s, 2s, and 2p. Inserting this expanded wavefunction into the TDSE generates a set of coupled differential equations of

$c_1(t)$, $c_2(t)$, and $c_3(t)$. The solution to these can be used to construct the dipole response of an electron, $d(t) = Re\{\sum_{i,j} d_{ij} c_i^*(t) c_j(t) e^{i(\epsilon_i - \epsilon_j)t}\}$; $i, j = 1, 2, 3$. $d_{ij}$ is the transition element between state $|i\rangle$ and $|j\rangle$. Before performing the Fourier transform of the dipole moment, a dephasing time was introduced by applying a time window function of 65 fs (lifetime of the 2p excited state) to preclude the possibility that the excited population induces a perpetual dipole moment. This three-level model has been used to investigate the sub-cycle oscillations of virtual states [28] and quantum interference [8,36], and it is appropriate for exploring the attosecond dynamics involving finite quantum states, such as those encountered in this study.

## 3. Results and Discussion

### 3.1. Basic Features of the ATA Spectrum below the Excited States

Figure 1 shows the ATA spectra of the helium atom calculated by scanning the time delay between the IR and XUV pulses using the 3D-TDSE (a) and three-level model (b). Negative (positive) delays indicate that the XUV pulse reaches the helium atom before (after) the IR pulse. The parameters of the XUV and IR pulses were described in Section 2. A significant observation is that only the tail of the frequency components of the XUV pulse covers the atomic absorption lines below the ionization potential, which is 24.58 eV under the SAE approximation. Therefore, the dynamics of electrons below the threshold, such as the half-cycle modulation of the 2p absorption line during the overlapping, the fringes around the 2p state, and the laser-induced states (LIS) of $2s^-$ are observable. These absorption features are fundamental to the ATA of helium atoms, which has been extensively studied in previous work. The denoted LIS follows the notation in [15]. For example, $2s^-$ indicates that the LIS is separated by one IR photon from the 2s state. Nevertheless, in contrast to previous results where the central photon energy of the XUV pulse was approximately 24 eV, a new absorption near $\omega_X = 14$ eV can be observed. During the overlapping (−2 O.C. to 2 O.C.) between the XUV and IR pulses, the absorption is half-cycle modulated along the delay axis, whereas at the large positive (>2 O.C.) or negative (<−2 O.C.) delays, the absorption spectrum is delay-independent with an absorption width relatively wider than that near the 2p state. It is worth pointing out that the phase control between the IR and the XUV pulse is crucial for the observed results. Due to the high quantum efficiency of the soft X-ray CCD camera, the spectrum of the XUV pulse with $10^{10}$ W/cm$^2$ can be experimentally detected.

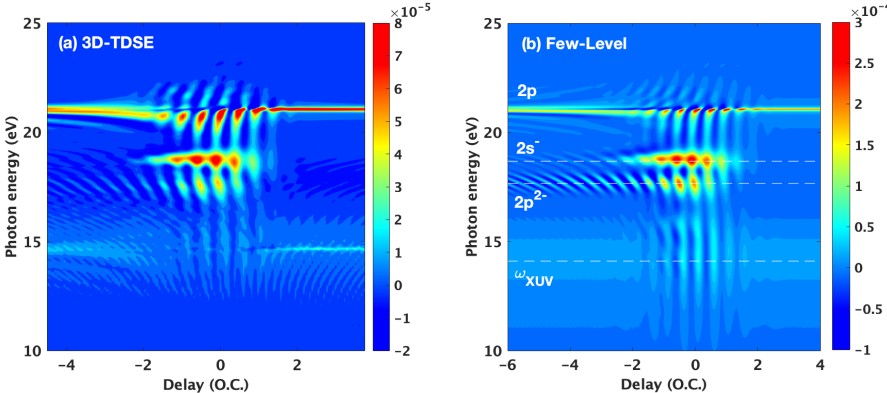

**Figure 1.** The ATA spectra (color coded) of the helium atom calculated from the response function $S(\omega, t_d)$ using the 3D-TDSE (**a**) and three-level model (**b**).

As previously demonstrated, the results of 3D-TDSE include all physical effects and its calculation is typically time-consuming. To capture the essential process of the electrons' dynamics shown in Figure 1a, the three-level model described in Section 2 was employed to simulate the ATA spectrum of helium atoms with the same parameters in Figure 1a. The results shown in Figure 1b generally recapture the features of Figure 1a; for instance, the

absorption of 2p, $2s^-$, $2p^{2-}$, and $\omega_X$ can be observed. The energy positions of 2p, $2s^-$, $2p^{2-}$, and $\omega_X$ are indicated by the white dashed lines.

To extract the delay-dependent information, the absorption signals for 2p, $2s^-$, and $\omega_X$ were calculated as the integrals of the response function around 2p (21.06 eV), $2s^-$ (18.67 eV), and $\omega_X$ (14 eV) with a spectral width of 0.4 eV. The three curves in Figure 2a present the absorption signals for 2p (blue), $2s^-$ (black), and $\omega_X$ (red). In the overlapping region, the absorption signals are distinctly modulated with a half-cycle period, and the absorption signals for $\omega_X$ exhibit behavior similar to that of 2p. Moreover, beyond the overlap phase, the absorption signals for $\omega_X$ exhibit time delay independence.

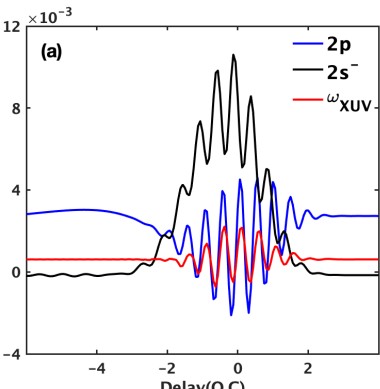 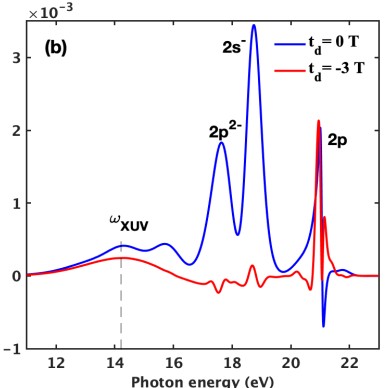

**Figure 2.** The delay-dependent absorption signals (**a**) and absorption spectrum (**b**) extracted from Figure 1b. The absorption signals of 2p, $2s^-$, and $\omega_X$ are represented by the blue, black, and red curves in Figure 2a respectively. Figure 2b is the absorption spectrum for two delays: blue curve for $t_d = 0T$ and red for $t_d = -3T$.

The half-cycle modulation of the absorption near $\omega_X$ is due to the phase induced by the IR field on the dipole generated by the XUV field [42]. This laser-imposed phase (LIP) depends on the IR field intensity; the absorption signals are delay-dependent, owing to their shared periodicity with the IR intensity. To verify this conclusion, the influence of the nearest 2s state is eliminated by artificially setting the transition element between 2s and 2p to 0. This is to establish whether the dark state 2s is coupled to the dipole induced by the XUV field. Even for $d_{1s-2p} = 0$ and other parameters remaining unchanged from Figure 1b, the three-level model ATA spectrum retains the half-cycle modulation for the absorption near $\omega_X$, indicating that the XUV-induced dipole and the dark state do not interact. Furthermore, on reducing the intensity of the IR field from $10^{13}$ W/cm$^2$ to $10^{10}$ W/cm$^2$, the modulation depth near $\omega_X$ is observed to be greatly weakened.

The wave function shows that the XUV pulse excites a wavepacket oscillating at the frequency of $\omega_X$ that is subsequently deformed by a delayed IR pulse. Meanwhile, during its motion in the IR pulse, the wavepacket acquires a phase proportional to the IR intensity, resulting in $2\omega_I$ modification along the delay axis, and $\omega_I$ is the angular frequency of the IR pulse.

Another unique feature, presented in Figure 1b, is that the width of the absorption spectrum near $\omega_X$ is much larger than that of 2p. To highlight the difference, the absorption spectrum was plotted for two typical delays: a blue curve for $t_d = 0T$ and a red curve for $t_d = -3T$, as shown in Figure 2b. The absorption near $\omega_X$, indicated by the black dashed line, for both delays, is gentler and wider than in other states, such as 2p, $2s^-$, and $2p^{2-}$. Furthermore, at zero delays, i.e., when the XUV and the IR pulses are overlapped, the presence of the IR field alters the absorption spectrum significantly.

### 3.2. Time-Dependent Perturbation Theory

The absorption near the center frequency of the XUV pulse $\omega_X$ is the major result of this study. We first investigated the absorption at large negative delays, where the IR field

can be ignored, i.e., only the XUV field interacting with the three-level system. The total Hamiltonian can be written as:

$$H = H_0 + V(t),  \tag{2}$$

where $H_0$ is the field-free Hamiltonian of the helium atom and $V(t)$ is the time-dependent perturbing Hamiltonian. In dipole approximation, $V(t) = -e\vec{r}E(t)$, where $e$ is the charge of electron, $\vec{r}$ is the position operator and $E(t)$ is the external electric field. Now, the wave function can be decomposed to a linear combination of the eigenbasis $|n\rangle$:

$$|\Psi(t)\rangle = \sum_n c_n(t)e^{-i\epsilon_n t/\hbar}|n\rangle  \tag{3}$$

where $\hbar$ is the reduced Planck constant, and $c_n(t)$ and $\epsilon_n$ are the time-dependent amplitude and eigen energy of state $|n\rangle$. By inserting Equation (3) into the time-dependent Schrödinger eqution, a set of coupled differential equations regarding the coefficients $c_n(t)$ can be obtained. The time-dependent perturbation is applied to solve the coupled equations, and for $q-$order term, the amplitude is:

$$c_n^{(q)}(t) = -\frac{i}{\hbar}\sum_k \int_0^t dt' \langle n|V(t')|k\rangle c_k^{(q-1)}(t')e^{-i(\epsilon_k - \epsilon_n)t'/\hbar}  \tag{4}$$

When the low-intensity XUV pulse interacts with a helium atom, only one XUV photon is absorbed. Therefore, the second-order approximation can accurately describe this process. Under this consideration, the maximum order applied here is $q = 2$. Therefore, the amplitude for state $|n\rangle$ is:

$$c_n(t) = c_n^{(0)}(t) + c_n^{(1)}(t) + c_n^{(2)}(t)  \tag{5}$$

As mentioned in the theoretical model Section, three eigen levels were used to simulate the absorption spectrum. $n = 1, 2, 3$ refer to the eigenstates 1s, 2s, and 2p. To simplify the calculation; the external electric field has a Gaussian formula:

$$E_{(I/X)}(t) = E_{(I/X)}\exp(-t^2/\tau_{(I/X)}^2)e^{i\omega_{(I/X)}t}  \tag{6}$$

where $I/X$ represents IR or XUV. $E_{(I/X)}$, $\tau_{(I/X)}$, and $\omega_{(I/X)}$ are the amplitude, duration, and center frequency of IR/XUV field. Iterated from the initial conditions $c_1^{(0)}(t) = 1$, $c_2^{(0)}(t) = 0$, and $c_3^{(0)}(t) = 0$, the wave function in Equation (3) can now be solved analytically:

$$
\begin{aligned}
|\Psi(t)\rangle = &\left\{e^{-i\frac{\epsilon_1}{\hbar}t} - \frac{e^2 E_X^2 d_{31}d_{13}e^{-2t^2/\tau_X^2}}{\hbar^2(\omega_X - \frac{\epsilon_1 - \epsilon_2}{\hbar})}\left[\frac{e^{i(2\omega_X - \frac{\epsilon_1}{\hbar})t}}{2\omega_X} - \frac{e^{i(\omega_X - \frac{\epsilon_3}{\hbar})t}}{\omega_X - \frac{\epsilon_3 - \epsilon_1}{\hbar}}\right]\right\}|1s\rangle \\
&- \frac{e^2 E_X^2 d_{23}d_{31}e^{-2t^2/\tau_X^2}}{\hbar^2(\omega_X - \frac{\epsilon_1 - \epsilon_3}{\hbar})}\left[\frac{e^{i(2\omega_X - \frac{\epsilon_1}{\hbar})t}}{2\omega_X - \frac{\epsilon_1 - \epsilon_2}{\hbar}} - \frac{e^{i(\omega_X - \frac{\epsilon_3}{\hbar})t}}{\omega_X - \frac{\epsilon_3 - \epsilon_2}{\hbar}}\right]|2s\rangle \\
&- \frac{ieE_X d_{31}e^{-t^2/\tau_X^2}}{\hbar}\frac{e^{i(\omega_X - \frac{\epsilon_1}{\hbar})t}}{\omega_X - \frac{\epsilon_1 - \epsilon_3}{\hbar}}|2p\rangle
\end{aligned}  \tag{7}
$$

where $d_{ij} = \langle i|\vec{r}|j\rangle$, $i, j = 1, 2, 3$ is the transition matrix for elements between states $|i\rangle$ and $|j\rangle$. This is the wave function for the electron interacting with a single XUV pulse. A lot of useful information can be directly extracted from Equation (7). The ground state $|1s\rangle$ majorly oscillates at the frequency of its eigen energy $\exp(-i\frac{\epsilon_1}{\hbar}t)$ and the XUV pulse simply induces a small perturbation to the ground state. The perturbation is proportional to the square of the XUV field, which is small in this study. The exponential term $\exp[i(\omega_X - \frac{\epsilon_1}{\hbar})t]$ presents the oscillation at the energy of $-\frac{\epsilon_1}{\hbar} + \omega_X$, corresponding to the absorption near $\omega_X$ observed in Figure 1a,b. Therefore, it can be concluded that an XUV pulse could excite a wavepacket oscillating at the frequency of the XUV pulse, even though the photon energy of

the XUV pulse is below the ionization potential ($\omega_X < I_p = 24.58$ eV). The time-dependent complex dipole near $\omega_X$ can be written as:

$$d_X(t) = -\frac{ieE_X d_{31}^2 e^{-t^2/\tau_X^2}}{\hbar(\omega_X - \frac{\epsilon_1 - \epsilon_3}{\hbar})} e^{i\omega_X t}, \tag{8}$$

The minus sign indicates the dipole near $\omega_X$ has a $\pi$ phase shift regarding the XUV field. The dipole spectrum can be obtained by making a Fourier transformation of Equation (8):

$$d_X(\Omega) = \int_{-\infty}^{+\infty} d_X(t)W(t)e^{-i\Omega t}dt \tag{9}$$

where $W(t)$ is the window function [15]:

$$W(t) = \cos^2 \frac{\pi t}{T_0} \tag{10}$$

The window function $W(t)$ introduces a dephasing time $T_0$ to inhibit the pertetual dipole moment. By integrating the Equation (9), the dipole spectrum can be obtained:

$$d_X(\Omega) = \frac{ie\sqrt{\pi}E_X \tau_X d_{31}^2}{\hbar(\omega_X - \frac{\epsilon_1 - \epsilon_3}{\hbar})} \left[ \frac{1}{2}e^{-\frac{1}{4}\tau_X^2(\Omega - \omega_X)^2} + \frac{1}{4}e^{-\frac{1}{4}\tau_X^2(\Omega + \frac{\pi}{T_0} - \omega_X)^2} + \frac{1}{4}e^{-\frac{1}{4}\tau_X^2(\Omega - \frac{\pi}{T_0} - \omega_X)^2} \right] \tag{11}$$

The spectrum of the XUV field can also be written as: $E_X(\Omega) = \sqrt{\pi}E_X \tau_X e^{-\frac{1}{4}\tau_X^2(\Omega - \omega_X)^2}$. Now the response function is:

$$S(\Omega) = \frac{e\pi E_X^2 \tau_X^2 d_{31}^2}{\hbar(\omega_X - \frac{\epsilon_1 - \epsilon_3}{\hbar})} e^{-\frac{1}{4}\tau_X^2(\Omega - \omega_X)^2} \left[ \frac{1}{2}e^{-\frac{1}{4}\tau_X^2(\Omega - \omega_X)^2} + \frac{1}{4}e^{-\frac{1}{4}\tau_X^2(\Omega + \frac{\pi}{T_0} - \omega_X)^2} + \frac{1}{4}e^{-\frac{1}{4}\tau_X^2(\Omega - \frac{\pi}{T_0} - \omega_X)^2} \right] \tag{12}$$

The response function $S(\Omega)$ in Equation (12) presents a Gaussian absorption profile, which is consistent with the simulation results in Figure 2b for $t_d = -3T$.

Now we turn our focus to the absorption during overlapping. Figure 1a,b shows that the absorption near $\omega_X$ is half-cycle modulated from $-2T$ to $+2T$. This $2\omega_I$ modulation is inevitably from the dressing effect of the IR pulse. During overlapping, the total electric field takes the formula:

$$E(t) = E_X \exp(-t^2/\tau_X^2)e^{i\omega_X t} + E_I \exp(-t^2/\tau_I^2)e^{i\omega_I t}e^{i\omega_I t_d} \tag{13}$$

where $\exp(i\omega_I t_d)$ is used to account the time delay between the IR and the XUV pulse. The $2\omega_I$ modulation indicates that two-IR photons are involved in the absorption process. Using the electric field in Equation (13), the time-dependent coefficients $c_1(t)$, $c_2(t)$, and $c_3(t)$ are:

$$\begin{aligned}
c_1(t) = &-\frac{e^2 d_{13}^2}{\hbar^2} \Bigg[ E_X^2 \frac{e^{i2\omega_X t} - 1}{2\omega_X(\omega_X - \frac{\epsilon_1 - \epsilon_3}{\hbar})} - E_X^2 \frac{e^{i(2\omega_X - \frac{\epsilon_3 - \epsilon_1}{\hbar})t} - 1}{\omega_X(\omega_X - \frac{\epsilon_1 - \epsilon_3}{\hbar})} \\
&+ E_X E_I \frac{e^{i(\omega_I - \frac{\epsilon_3 - \epsilon_1}{\hbar})t} - 1}{(\omega_X - \frac{\epsilon_1 - \epsilon_3}{\hbar})(\omega_X - \frac{\epsilon_1 - \epsilon_3}{\hbar})} e^{i\omega_I t_d} - E_X E_I \frac{e^{i(\omega_X - \frac{\epsilon_3 - \epsilon_1}{\hbar})t} - 1}{(\omega_I - \frac{\epsilon_1 - \epsilon_3}{\hbar})(\omega_X - \frac{\epsilon_1 - \epsilon_3}{\hbar})} e^{i\omega_I t_d} \\
&+ E_X E_I \frac{e^{i(\omega_X + \omega_I)t} - 1}{(\omega_X - \frac{\epsilon_1 - \epsilon_3}{\hbar})(\omega_X + \omega_I)} e^{i\omega_I t_d} - E_X E_I \frac{e^{i(\omega_I - \frac{\epsilon_3 - \epsilon_1}{\hbar})t} - 1}{(\omega_I - \frac{\epsilon_1 - \epsilon_3}{\hbar})(\omega_X - \frac{\epsilon_1 - \epsilon_3}{\hbar})} e^{i\omega_I t_d} \\
&+ E_I^2 \frac{e^{i2\omega_I t} - 1}{2\omega_I(\omega_I - \frac{\epsilon_1 - \epsilon_3}{\hbar})} e^{i2\omega_I t_d} - E_I^2 \frac{e^{i(2\omega_I - \frac{\epsilon_3 - \epsilon_1}{\hbar})t} - 1}{(\omega_I - \frac{\epsilon_3 - \epsilon_1}{\hbar})(\omega_I - \frac{\epsilon_1 - \epsilon_3}{\hbar})} e^{i2\omega_I t_d} \Bigg]
\end{aligned} \tag{14}$$

$$c_2(t) = -\frac{e^2 d_{23} d_{31}}{\hbar^2} \Bigg[ E_X^2 \frac{e^{i(2\omega_X + \frac{\epsilon_2 - \epsilon_1}{\hbar})t} - 1}{(2\omega_X - \frac{\epsilon_1 - \epsilon_3}{\hbar})(\omega_X - \frac{\epsilon_1 - \epsilon_2}{\hbar})} - E_X^2 \frac{e^{i(\omega_X - \frac{\epsilon_3 - \epsilon_2}{\hbar})t} - 1}{(\omega_X - \frac{\epsilon_3 - \epsilon_2}{\hbar})(\omega_X - \frac{\epsilon_1 - \epsilon_3}{\hbar})}$$

$$+ E_X E_I \frac{e^{i(\omega_X + \omega_I - \frac{\epsilon_1 - \epsilon_2}{\hbar})t} - 1}{(\omega_I - \frac{\epsilon_1 - \epsilon_3}{\hbar})(\omega_X + \omega_I - \frac{\epsilon_1 - \epsilon_2}{\hbar})} e^{i\omega_I t_d} - E_X E_I \frac{e^{i(\omega_X - \frac{\epsilon_3 - \epsilon_2}{\hbar})t} - 1}{(\omega_I - \frac{\epsilon_1 - \epsilon_3}{\hbar})(\omega_X - \frac{\epsilon_3 - \epsilon_2}{\hbar})} e^{i\omega_I t_d}$$

$$+ E_X E_I \frac{e^{i(\omega_I + \omega_X - \frac{\epsilon_1 - \epsilon_2}{\hbar})t} - 1}{(\omega_X - \frac{\epsilon_1 - \epsilon_3}{\hbar})(\omega_X + \omega_I - \frac{\epsilon_1 - \epsilon_2}{\hbar})} e^{i\omega_I t_d} - E_X E_I \frac{e^{i(\omega_I - \frac{\epsilon_3 - \epsilon_2}{\hbar})t} - 1}{(\omega_X - \frac{\epsilon_1 - \epsilon_3}{\hbar})(\omega_I - \frac{\epsilon_3 - \epsilon_2}{\hbar})} e^{i\omega_I t_d}$$

$$+ E_I^2 \frac{e^{i(2\omega_I - \frac{\epsilon_1 - \epsilon_2}{\hbar})t} - 1}{(\omega_I - \frac{\epsilon_1 - \epsilon_3}{\hbar})(2\omega_I - \frac{\epsilon_1 - \epsilon_2}{\hbar})} e^{i2\omega_I t_d} - E_I^2 \frac{e^{i(\omega_I - \frac{\epsilon_3 - \epsilon_2}{\hbar})t} - 1}{(\omega_I - \frac{\epsilon_1 - \epsilon_3}{\hbar})(\omega_I - \frac{\epsilon_3 - \epsilon_1}{\hbar})} e^{i2\omega_I t_d} \Bigg] \quad (15)$$

$$c_3(t) = -\frac{ie E_X d_{31}}{\hbar} \frac{e^{i(\omega_X - \frac{\epsilon_1 - \epsilon_3}{\hbar})t}}{\omega_X - \frac{\epsilon_1 - \epsilon_3}{\hbar}} - \frac{ie E_I d_{31}}{\hbar} \frac{e^{i(\omega_I - \frac{\epsilon_1 - \epsilon_3}{\hbar})t}}{\omega_I - \frac{\epsilon_1 - \epsilon_3}{\hbar}} e^{i\omega_I t_d} \quad (16)$$

It should be noticed that the envelope of the XUV and IR field has been ignored for simplicity. Similarly to Equation (7), the wave function of electrons interacting with the combined fields XUV+IR can also be written down.

Focusing on the absorption near $\omega_X$, the relevant complex dipole can now be written as:

$$d_X(t) = -\frac{ie^3 d_{31}^2 E_X E_I^2 e^{i\omega_X t} e^{i2\omega_I t_d}}{\hbar^3 (\omega_I - \frac{\epsilon_1 - \epsilon_3}{\hbar})(\omega_X - \frac{\epsilon_1 - \epsilon_3}{\hbar})} \Bigg[ \frac{d_{13}}{2\omega_I} - \frac{d_{13}}{\omega_I - \frac{\epsilon_3 - \epsilon_1}{\hbar}} + \frac{d_{23}}{2\omega_I - \frac{\epsilon_1 - \epsilon_2}{\hbar}} - \frac{d_{23}}{\omega_I - \frac{\epsilon_3 - \epsilon_2}{\hbar}} \Bigg] \quad (17)$$

This time- and delay-dependent dipole reveals the dressing effect of the IR field on the three-level atom: the dipole is modulated with a period of $2\omega_I$. The structure of Equation (17) also proves the discussions regarding the absence of the 2s state not removing the $2\omega_I$ modulation near $\omega_X$: setting $d_{23} = 0$ does not affect the $d_{13}$ couplings. Specifically, the response function during the overlapping is:

$$S(\Omega, t_d) = [E_X'(\Omega) + \frac{1}{2} E_X'(\Omega - \frac{\pi}{T_0}) + \frac{1}{2} E_X'(\Omega + \frac{\pi}{T_0})] E_X(\Omega) \sin(2\omega_I t_d), \quad (18)$$

where $E_X'(\Omega)$ is the Fourier spectrum of the dipole in Equation (17) ignoring the delay-dependent term, and $E_X'(\Omega \pm \frac{\pi}{T_0})$ comes from the time window function, similar to the results in Equation (12). The response function in Equation (16) explains the half-cycle modulation of the absorption spectrum in Figure 1a,b. In addition, by observing the structure of the time-dependent coefficients of $c_1(t)$, $c_2(t)$, and $c_3(t)$ in Equations (14)–(16), one may find the delay-dependent term of $\exp(i\omega_I t_d)$, which may lead to the $\omega_I$ modulation of the absorption spectrum along the delay axis. However, this $\omega_I$ modulation seems absent in Figure 1a,b. The reason for the absence of the $\omega_I$ modulation is that the $2\omega_I$ modulation dominates the absorption process, and the $\omega_I$ modulation period can be identified by performing a Fourier transform of the spectrum in Figure 1b along the delay axis, as shown in Figure 3.

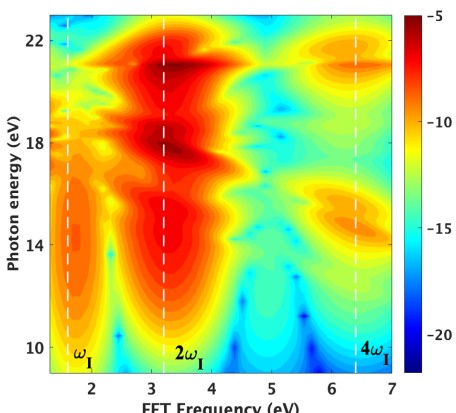

**Figure 3.** Fourier transformation of the absorption spectrum in Figure 1b along the delay axis.

Figure 3 is the Fourier transformation along the delay axis for each XUV frequency. The white-dashed lines indicate the positions of $\omega_I$, $2\omega_I$, and $4\omega_I$ modulation. The log scale has been applied to display the weak $\omega_I$ and $4\omega_I$ components. The presence of the $\omega_I$ modulation proves our previous discussion regarding the time-dependent coefficients $c_1(t)$, $c_2(t)$, and $c_3(t)$. The $4\omega_I$ modulation indicates four IR photons are involved in the absorption, which can not be revealed by the perturbation method used in Equations (14)–(16).

After completing the perturbated illustration of the absorption near $\omega_X$, the three-level model is considered to investigate the chirp effect of the XUV pulse. In Figure 4, the ATA spectrum of the helium atom was calculated for two XUV chirp: (a) $\xi = -3$ and (b) $\xi = +3$. The delay-dependent feature is observed during overlapping when the absorption fringes differ in their inclination for different XUV chirps (as indicated by the black dashed line in Figure 4a). By the chirped formula of the XUV field expressed by Equation (1), the central XUV pulse frequency shifts with varying XUV chirp, as shown in Figure 4a,b. However, the XUV chirp effect on the ATA spectrum remains unchanged despite the shift. The slope of the fringes is a single-valued function of the chirp of the XUV pulse, which was confirmed by scanning the chirp from $-3$ to $+3$ in steps of 0.2 (not shown here).

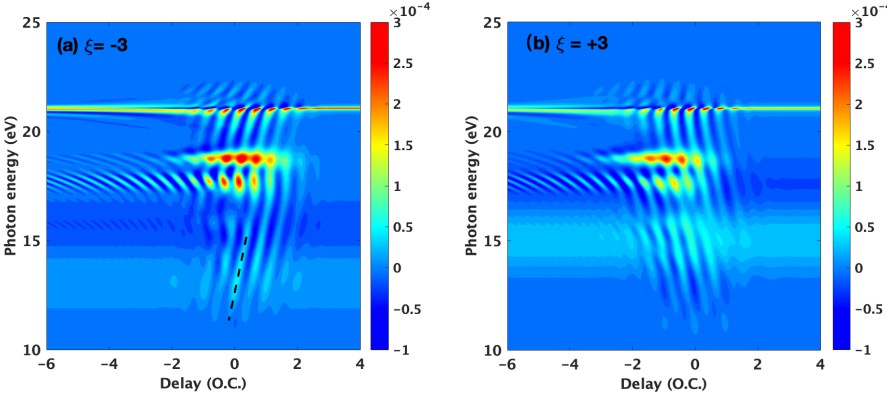

**Figure 4.** Absorption spectrum of helium atom obtained by calculating the response functions for two chirps: (**a**) $\xi = -3$ and (**b**) $\xi = +3$.

A potential application of the chirp-dependent ATA spectrum shown in Figure 4 is as an all-optical method to extract the time duration of the chirped XUV pulse. The chirp of the XUV pulse can be extracted from the slope of the absorption fringes: $\xi = \xi(s)$, where $s$ is the slope, following which the XUV pulse duration can be estimated as $\tau = \tau_0 \sqrt{1 + \xi^2(s)}$ [43] and $\tau_0$ is the Fourier transform-limited duration. It is noteworthy that the absorption near $\omega_X$ is notably weak because the spectrum components within its bandwidth do not resonate with any atomic intrinsic states. The weak absorption made it difficult to calculate

the results in the experiment. However, considering the propagation effect, the weak absorption near $\omega_X$ may be enhanced by applying a helium gas at high pressure.

## 4. Conclusions

In this study, the ATA spectrum of the helium atom was theoretically and numerically investigated under the SAE approximation. The ATA spectrum was obtained as the response function derived from the energy exchange between the electron and the total electric field. A fully 3D-TDSE and a three-level model numerically calculated the response function. However, the central XUV frequency used in the simulation was 14 eV, which is less than the energy difference between atomic states 1s and 2p. Therefore, only the tail components of the XUV spectrum can excite electrons from the ground state to the excited state through single-photon resonant absorption. Nevertheless, the results of the 3D-TDSE and three-level model indicated that the central frequency components of such XUV pulses can also induce absorption that is significantly wider near $\omega_X$ than near 2p. During overlapping, the absorption near $\omega_X$ is half-cycle modulated, whereas at large delays (positive or negative), the absorption is delay-independent. Analysis using the time-dependent perturbation theory demonstrated that the absorption near $\omega_X$ corresponds to the wavepacket excited by the XUV field.

In addition, the three-level model shows that the absorption fringes near $\omega_X$ are also dependent on the chirp of the XUV pulse. Most importantly, the slope of these fringes is a single-valued function of the XUV chirp, which might provide an all-optical method for characterizing the XUV pulse duration.

**Author Contributions:** Conceptualization, J.X. and M.W.; software, J.X.; formal analysis, J.X. and M.W.; writing—original draft preparation, J.X. and X.W.; supervision, C.Z.; funding acquisition, M.W., C.Z. and S.R. All authors have read and agreed to the published version of the manuscript.

**Funding:** This work is funded by the National Key R&D Program of China (grant number 2016YFA0401100), the National Natural Science Foundation of China (grants 11875092, 11575031, and 12005149), the Featured Innovation Project of Educational Commission of Guangdong Province of China (grant number 2018KTSCX352), the Natural Science Foundation of Top Talent of SZTU (grants 2019010801001 and 2019020801001), and Natural Science Basic Research Program of Shaanxi (program number 2020JQ-204).

**Institutional Review Board Statement:** Not applicable.

**Informed Consent Statement:** Not applicable.

**Data Availability Statement:** The data and simulation code involved in this study can be obtained from the authors.

**Acknowledgments:** The authors of this paper acknowledge helpful discussions with Peng Peng from ShanghaiTech University.

**Conflicts of Interest:** The authors declare no conflict of interest.

## Abbreviations

The following abbreviations are used in this manuscript:

| | |
|---|---|
| ATA | Attosecond transient absorption |
| XUV | Extreme ultraviolet |
| IR | Infrared |
| ATAS | Attosecond transient absorption spectroscopy |
| 3D-TDSE | Three-dimentional time-dependent Schrödinger equation |
| SAE | Single-active electron |
| O.C. | Optical cycle |
| LIS | Laser-induced state |

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
