# Peer review of "Attosecond Transient Absorption Below the Excited States"

_photonics, doi:10.3390/photonics9040269_

Round 1
Reviewer 1 Report
The manuscript ID photonics-1650495 entitled “Attosecond Transient Absorption Below the Excited States,” Jinxing Xue, etc.
As far as the understanding of this Referee has, this manuscript shows a very good theoretical study of ATA in Helium through ultrafast sciences pulses. The presentation, the logic, and the research question behind the mathematical development are “right” and beautiful.
Before clarifying a few points, I am happy to recommend this manuscript for publication at MPDI.
1) I wondered in Fig. 1 where the difference between Fig1a) and Fig1b) come?
2) Can the Authors explain the Floquet picture from a mathematical point of view?
3) In the introduction, if the authors are considering Ref[12] is meaningful, the same citations should be given to the papers:
Phys. Rev. B 102, 134115 (2020) and ACS Nano Lett., 21, 8970–8978 (2021) (https://doi.org/10.1021/acs.nanolett.1c02145)
Am I right?
Thanks for your good job.
After these modifications, I recommend the paper for publication at MPDI.
Reviewer 2 Report
It is well-known that an external constant or slowly-varying electric field can make a medium absorb light in a spectral range where there are no transitions that would allow for absorption in the absence of the field. Let us call that external field a dressing field and the pulse that probes the induced absorption a probe pulse. When the dressing field is not too strong and its frequency is not too small, we have a usual multiphoton process. In this process, the simultaneous absorption of a probe-pulse photon and at least one photon from the dressing pulse enable transitions that would otherwise be energetically forbidden. Alternatively, the induced absorption can be interpreted as the dressing field Stark-shifting energy levels.
Even though the effect is well-known, it has not been thoroughly investigated through attosecond absorption spectroscopy. This is what the manuscript that I am reviewing does. As their main approach, the authors have chosen the Floquet theory, which was done before, but previous studies considered Floquet states induced by the dressing field, while this manuscript utilizes Floquet states induced by the probe field. Since the Floquet approach usually becomes useless for few-cycle pulses, it is uncertain whether it should work for the two-cycle probe pulse employed in this manuscript. The manuscript looks interesting to me, but, having read it, I am not sure how well the Floquet approach actually works here. I also think that the authors could do a much better job explaining their numerical results in the framework of the Floquet formalism.
What disturbs me most is that the authors do not explicitly relate the "response function" (ATA spectra, absorption spectra) to their Floquet states. Because of this, the reader is left uncertain about what statements like these mean: "The absorption observed around ω_XUV is shown in Figs. 1(a) and 1(b), can be attributed to the absorption occurring in Floquet state |1s, +1⟩ ..." and "As the absorption near ω_XUV is attributed to the Floquet state |1s, +1> ..." To begin with, absorption is never attributed to a single state--it is always attributed to a transition from one state to another one. Second, that Floquet state also exists in the absence of the dressing field, but there would obviously be no absorption at ω_XUV without the dressing field. So, the word "attributed" in the above quotes is extremely vague. Thinking about the underlying physics, I realize that the XUV pulse adiabatically populates the |1s, +1> state, while the infrared pulse drives the transition from |1s, +1> to |2s, m> and |2p, m> states, but
- I am not sure which of the final states matters most.
- It is not obvious to me how those infrared-driven transitions make the dipole moment oscillate at ω_XUV with a phase shift that corresponds to absorption.
The authors must clarify these things. As far as I can judge, it is best to clarify them by deriving an expression for the response function in the basis of their Floquet states.
Another problem that I see is that Eq. (6) is rather guessed than rigorously derived. I understand that this is a well-known expression from the second-order perturbation theory, but it is not obvious that it applies here. In fact, introducing this equation, the authors cite [40], but this paper does not even contain the word "Floquet". In the formalism where Floquet states are induced by the XUV (probe) pulse, the IR-induced absorption is due to transitions driven by the IR field between the Floquet states. That equation in [40] was derived for a very different case, and it is not clear whether it applies here. If it does, then a rigorous derivation is due.
In addition, I have a few minor comments:
- The authors should specify how the light pulses were polarized. (I suppose that they were linearly polarized and that their fields were parallel, but this is not written anywhere.)
- The authors should not use the same symbol ($\varepsilon$) for the electric field and the Floquet energy.
- The LIS abbreviation must be spelt out where it is used for the first time.
Reviewer 3 Report
See report

Round 2
Reviewer 2 Report
This revision is a significant improvement, but there is at least one thing that I find very confusing, if not wrong. The response function is defined as the imaginary part of the product where the Fourier transform of the dipole moment is multiplied with the complex-conjugate Fourier transform of the XUV pulse. However, evaluating the response function on page 6, the authors first take the Fourier transform of the imaginary part of the dipole moment, see Eq. (9). I don't see how this could be right.
A few minor comments:
1) I think that "for the first time" should be erased from the abstract. Absorption of XUV light at frequencies below the 1s-2p transition was simulated before [see, e.g., Wu et al., J. Phys. B: At. Mol. Opt. Phys. 49 (2016) 062003].
2) On line 34, "electron" should be replaced with "electronic".
3) "ban be absorbed" should be replaced with "can be absorbed"; that entire sentence needs to be revised.
Reviewer 3 Report
It seems to me that when XUV intensity increases, it is inevitable that the continuous spectrum will play a role. Of course, when the basis set excludes it, this fact is not revealed.
A paper of relevance might be, Mercouris et al J Phys B 53, 095603 (2020)
Author Response
Thanks. The paper: Mercouris et al J Phys B 53, 095603 (2020) is helpful, we have added it as Ref. [23] in the revised manuscript.